# A novel score system of blood tests for differentiating Kawasaki disease from febrile children

**Chih-Min Tsai**[1], **Chi-Hsiang Chu**[2], **Xi Liu**[3], **Ken-Pen Weng**[4,5], **Shih-Feng Liu**[6,7], **Ying-Hsien Huang**[1,8]*, **Ho-Chang Kuo**[1,6,8]*

**1** Department of Pediatrics, Kaohsiung Chang Gung Memorial Hospital and Chang Gung University College of Medicine, Kaohsiung, Taiwan, **2** Department of Statistics, National Cheng Kung University, Tainan, Taiwan, **3** Department of Pediatrics, Baoan Women's and Children's Hospital, Jinan University, Shenzhen, China, **4** Department of Pediatrics, Kaohsiung Veterans General Hospital, Kaohsiung, Taiwan, **5** Faculty of Medicine, National Yang-Ming University, Taipei, Taiwan, **6** Department of Respiratory Therapy, Kaohsiung Chang Gung Memorial Hospital, Kaohsiung, Taiwan, **7** Department of Internal Medicine, Division of Pulmonary and Critical Care Medicine, Kaohsiung Chang Gung Memorial Hospital, and Chang Gung University College of Medicine, Kaohsiung, Taiwan, **8** Kawasaki Disease Center, Kaohsiung Chang Gung Memorial Hospital, Kaohsiung, Taiwan

☯ These authors contributed equally to this work.

\* yhhuang123@yahoo.com.tw (YHH); erickuo48@yahoo.com.tw (HCK)

**Data Availability Statement:** All relevant data are within the paper and its Supporting Information files.

**Funding:** This study received funding from the following grants: MOST 108-2314-B-182 -037

## Abstract

### Background

Kawasaki disease is the most common cause of acquired heart disease among febrile children under the age of 5 years old. It is also a clinically diagnosed disease. In this study, we developed and assessed a novel score system using objective parameters to differentiate Kawasaki disease from febrile children.

### Methods

We analyzed 6,310 febrile children and 485 Kawasaki disease subjects in this study. We collected biological parameters of a routine blood test, including complete blood count with differential, C-reactive protein, aspartate aminotransferase, and alanine aminotransferase. Receiver operating characteristic curve, logistic regression, and Youden's index were all used to develop the prediction model. Two other independent cohorts from different hospitals were used for verification.

### Results

We obtained eight independent predictors (platelets, eosinophil, alanine aminotransferase, C-reactive protein, hemoglobin, mean corpuscular hemoglobin, mean corpuscular hemoglobin concentration, and monocyte) and found the top three scores to be eosinophil >1.5% (score: 7), alanine aminotransferase >30 U/L (score: 6), and C-reactive protein>25 mg/L (score: 6). A score of 14 represents the best sensitivity value plus specificity prediction rate for Kawasaki disease. The sensitivity, specificity, and accuracy for our cohort were 0.824, 0.839, and 0.838, respectively. The verification test of two independent cohorts of Kawasaki

-MY3 and MOST 103-2410-H-264-004 from the Ministry of Science and Technology of Taiwan and 8E0212 from Chang Gung Memorial Hospital, Taiwan. Although these institutes provided financial support, they had no influence on the way in which we collected, analyzed, or interpreted the data or wrote this manuscript.

**Competing interests:** The authors have declared that no competing interests exist.

disease patients (N = 103 and 170) from two different institutes had a sensitivity of 0.780 (213/273).

## Conclusion

Our findings demonstrate a novel score system with good discriminatory ability for differentiating between children with Kawasaki disease and other febrile children, as well as highlight the importance of eosinophil in Kawasaki disease. Using this novel score system can help first-line physicians diagnose and then treat Kawasaki disease early.

## Introduction

The most common reason that young children seek medical care in pediatric emergency departments (PED) is fever [1]. In most cases, the primary causes of such fever consist of self-limited viral infection or bacterial infection [2]. Among febrile children (FC), Kawasaki disease (KD) is the most concerning with regard to acquired heart disease in children under the age of 5 years old [3]. KD is characterized by prolonged fever for more than 5 days, bilateral non-purulent conjunctivitis, diffuse mucosal inflammation, polymorphous skin rashes, indurative angioedema of the hands and feet, and non-suppurative cervical lymphadenopathy; fever lasting more than 96 hours with erythematous changes on the palm and three other major symptoms can also fit the diagnosis of KD [4, 5]. However, all the major signs for diagnosing KD are subjective. To prevent the most severe sequela or complications of coronary artery aneurysms (CAA) in KD patients [4, 6], intravenous immunoglobulin (IVIG) therapy needs to be introduced early in order to reduce the complication of CAA. Both randomized, controlled studies and meta-analyses have confirmed that IVIG treatment in KD patients is most effective when given within 10 days of fever onset and also reduces the risk of CAA [5, 7–9]. Therefore, early awareness of KD is particularly important for both clinicians and parents. However, the greatest challenge for clinicians in PED is the early identification of KD itself because KD shares many clinical signs with other childhood febrile illnesses [10]. Furthermore, 20%-30% of KD patients do not completely fulfill the above diagnostic criteria and are thus considered as incomplete KD, which often makes diagnosis even more challenging for clinicians that are not experienced pediatricians [5, 11].

Although no single laboratory examination can serve as a golden standard diagnostic tool for KD, we have endeavored to develop many quick methods for the early identification of KD [12–14]. However, such methods are still far from being applied in clinical practice. Ling et al. described a KD scoring system with either clinical presentation, laboratory test results, or their combination to differentiate KD from FC [15] and further improved their two-step algorithm with another cohort in 2016 [10]. Nevertheless, subjective clinical presentations still played an important role in their prediction model. In this study, we developed and assessed a new scoring system of objective routine blood measurements without any clinical inspection data for distinguishing KD in young FC in PED.

## Materials and methods

### Study population

This study is a retrospective case-control study. Both complete and incomplete KD patients diagnosed in Kaohsiung Chang Gung Memorial Hospital from January 2009 to December

2017 were enrolled to form our case group. The clinical data sources of the KD group were accessed by Prof. H-C Kuo. Both complete and incomplete KD patients were required to meet the diagnosis criteria stipulated by the American Heart Association [5]. The diagnosis of complete KD is based on the presence of ≥ 5 days of fever and the presence of ≥ 4 of the 5 principal clinical features. These five features consist of the following: (1) erythema and cracking of lips, strawberry tongue, and/or erythema of oral and pharyngeal mucosa, (2) bilateral bulbar conjunctival injection without exudate, (3) rash: maculopapular, diffuse erythroderma, or erythema multiforme-like, (4) erythema and edema of the hands and feet in the acute phase and/or periungual desquamation in the subacute phase, and (5) cervical lymphadenopathy (≥1.5 cm diameter), usually unilateral. In the presence of ≥ 4 principal clinical criteria, particularly when redness and swelling of the hands and feet are present, a KD diagnosis may be made with only 4 days of fever. Children with fever ≥ 5 days and 2 or 3 compatible clinical criteria are then evaluated with additional laboratory tests. If these children had ≥3 supplemental laboratory criteria (anemia for age, platelet count ≥450,000 after the seventh day of fever, albumin ≤3.0 g/dL, elevated ALT level, WBC count ≥15,000/mm3, and ≥10 WBC/HPF on urinalysis) or a positive echocardiogram, a diagnosis of incomplete KD was made. In our KD series, we have demonstrated that 94% of patients diagnosed with KD were under the age of 5 years old [16]. Therefore, we enrolled patients younger than five years old without a diagnosis of KD as our control group. These controls presented to the PED of Kaohsiung Chang Gung Memorial Hospital, Taiwan with fever during the period of January 2009 through December 2015. Children suspected of having KD were excluded from the FC group. The clinical data sources of the control group were accessed by Prof. Y-H Huang and Dr. C-M Tsai. We collected data of demographic characteristics (age and gender) and biological parameters, including complete blood count with differential count (CBC/DC), C-reactive protein (CRP), aspartate aminotransferase (AST), and alanine aminotransferase (ALT). We did not include erythrocyte sedimentation rate, albumin, and urinalysis, which are among the laboratory testing for KD suggested by the AHA [5] because they were not routinely tested in FC at our PED. This study was approved by the Chang Gung Medical Foundation's Institutional Review Board (IRB number:201801163B0).

## Statistical analysis

For continuous variables, we adopted independent t-test or Mann-Whitney U test to compare differences between the KD and FC groups with correspondence to the normality test. We applied the chi-squared test (or Fisher's exact test) to analyze the discrete variables. The receiver operating characteristic (ROC) curve was used to calculate the cutoff values in different variables to predict KD. The best possible cut-off value was indicated by the highest Youden's Index. Logistic regression models with univariate and multivariate analyses were performed to identify potential independent predictors for KD. We carried out statistical analysis using SPSS statistical software for Windows version 22 (SPSS for Windows, version 22; SPSS, IL, U.S.A.). A value of $p < 0.05$ was considered statistically significant.

## Analysis process

We enrolled patients with complete data to construct the model. First, we used the ROC curve to select the candidate variables from the biological parameters with significance of area under the curve (AUC) greater than 0.6 or less than 0.4. Then an appropriate cutoff point of each candidate variable was calculated using Youden's index. Second, we divided data into two disjointed data sets: the modeling set contained 83.3% (5/6) of all cases and the testing set contained 16.7% (1/6) of all cases in order to construct the model and validate it by randomization

(uniform random and sorting with MATLAB 2015b), respectively. To construct the model, we used the logistic regression model with univariate and multivariate analyses to identify potential independent predictors and then generated the candidate score system based on the estimated coefficients, which we rounded off. We randomly performed five-fold cross-validation to determine the best score system with the highest prediction rate (sensitivity plus specificity). The cutoff point of the score system was also calculated using the ROC curve and Youden's index. Fig 1 shows the workflow for analysis. We also used two other independent KD cohorts from different hospitals (Baoan Maternal and Child Health Hospital in China, N = 170, and Kaohsiung Veterans General Hospital in Taiwan, N = 103, data accessed by X Liu and K-P Weng, respectively, and with the same period of our KD group of 2009 to 2017) for final verification.

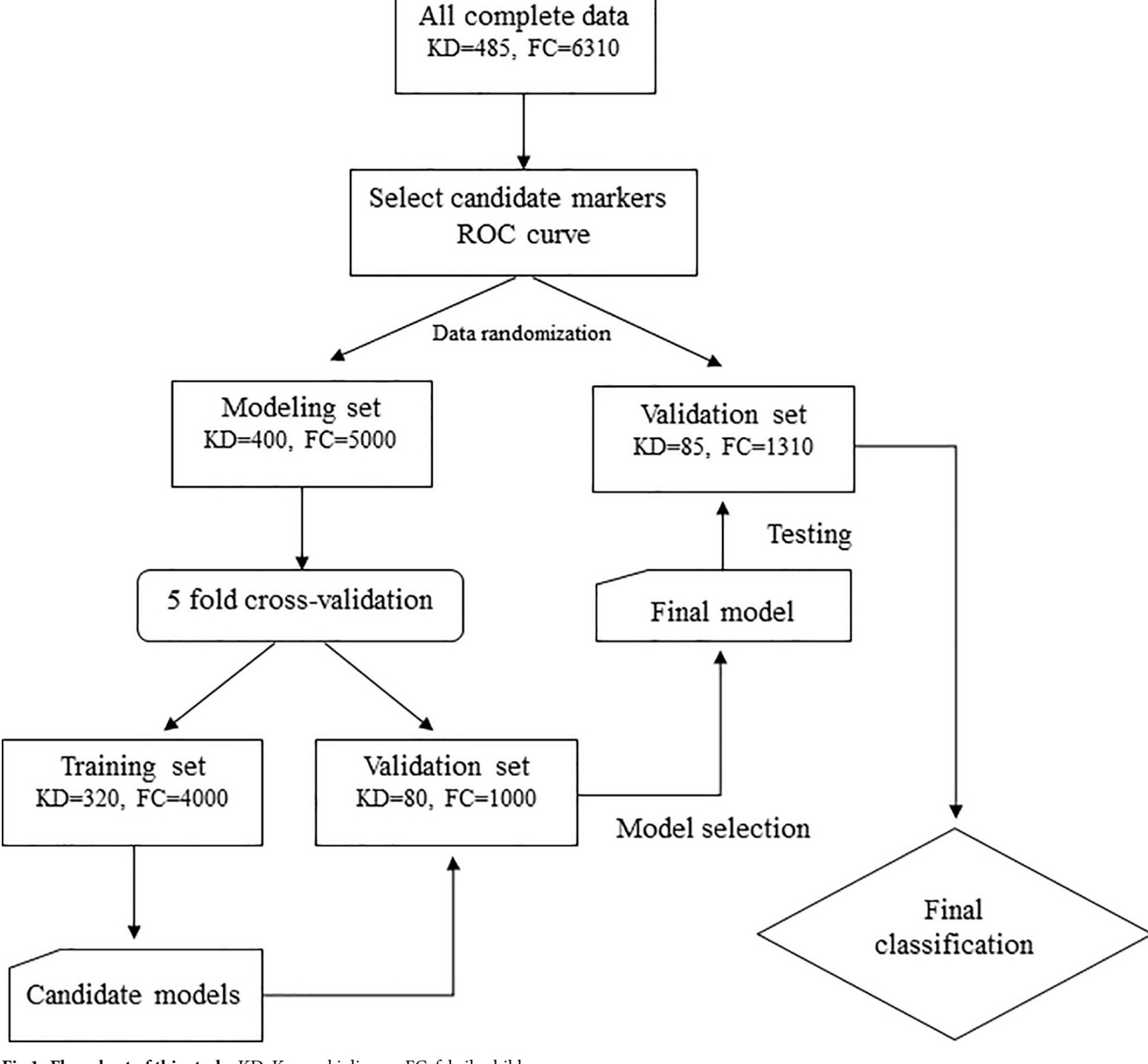

**Fig 1. Flow chart of this study.** KD: Kawasaki disease, FC: febrile children.

## Results

### Patients' characteristics

In this study, we enrolled 6,310 FC and 485 KD patients, including 358 complete KD and 127 incomplete KD. Among these 6,795 participants, 400 KD and 5,000 FC formed our modeling set (N = 5,400) and 85 KD and 1,310 FC formed our validation set (N = 1,395). The basic demographics and biological data in both groups are listed in Table 1. Except for the blood parameter of AST, we observed significant differences between KD and FC in complete blood count, ALT, and CRP.

### Parameters for distinguishing KD and FC

Fig 2 and Table 2 show the ROC curve plot and the selected candidate parameters with AUC and the appropriate cutoff points, respectively. We initially obtained the five increased parameters of white blood cells, platelets, eosinophil, ALT, and CRP, as well as the five decreased parameters of red blood cell, hemoglobin, MCH, MCHC, and monocyte, between KD and FC. Thereafter, we used a logistic regression model with univariate and multivariate analyses to identify the important predictors for KD. We then gave each significant predictor an individual score based on the estimated coefficients with rounding off (Table 3). In the end, we obtained these eight independent predictors (platelets, eosinophil, ALT, CRP, hemoglobin, MCH, MCHC, and monocyte) for the logistic regression model to generate the candidate score system. In this score system, the top three scores were eosinophil>1.5% (7), ALT>30 U/L (6), and CRP>25 mg/L (6).

### Performance of our classification model

We randomly performed five-fold cross-validation to determine the best score system with the greatest prediction rate (sensitivity plus specificity). The cutoff point of the score system was

**Table 1. Basic characteristics between controls and Kawasaki disease.**

| Variable | Controls (N = 6310) | Kawasaki disease (N = 485) | Effect size* | p-value |
|---|---|---|---|---|
| Gender (male) | 3467 (54.9%) | 285 (58.8%) | 0.095 | 0.103 |
| Age (year) | 1.96±1.36 | 1.91±1.63 | 0.036 | 0.001 |
| WBC (1000/μL) | 11.36±5.94 | 13.17±4.67 | 0.309 | <0.001 |
| RBC (million/μL) | 4.42±0.48 | 4.28±0.43 | 0.294 | <0.001 |
| Hemoglobin (g/dL) | 11.94±1.24 | 11.33±5.46 | 0.324 | <0.001 |
| MCV (fL) | 80.20±5.41 | 78.28±5.44 | 0.355 | <0.001 |
| MCH (pg/cell) | 27.13±2.02 | 26.06±2.26 | 0.525 | <0.001 |
| MCHC (gHb/dL) | 33.81±0.89 | 33.20±1.88 | 0.614 | <0.001 |
| Platelets (1000/μL) | 273.52±105.92 | 353.43±138.49 | 0.736 | <0.001 |
| Segment (%) | 55.09±18.00 | 57.62±16.10 | 0.142 | 0.004 |
| Lymphocyte (%) | 34.19±16.62 | 31.71±14.78 | 0.150 | 0.004 |
| Monocyte (%) | 8.78±4.03 | 6.08±2.96 | 0.681 | <0.001 |
| Eosinophil (%) | 0.95±1.43 | 3.13±3.13 | 1.353 | <0.001 |
| Basophil (%) | 0.25±0.32 | 0.22±0.32 | 0.094 | 0.001 |
| CRP (mg/L) | 35.11±52.22 | 77.88±68.04 | 0.799 | <0.001 |
| AST (U/L) | 41.11±42.10 | 65.26±85.88 | 0.518 | 0.805 |
| ALT (U/L) | 23.48±33.33 | 74.80±104.34 | 1.207 | <0.001 |

* Effect size: Cohen's *h* for category variables, Cohen's *d* for continuous variables.

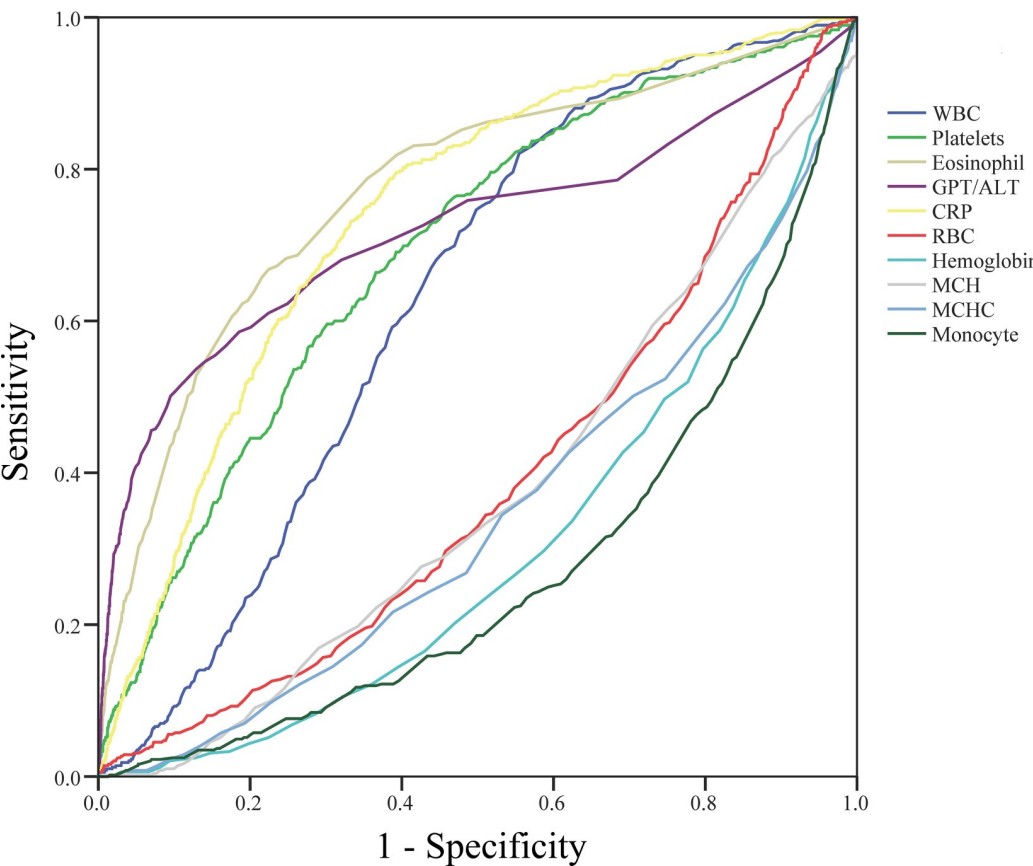

**Fig 2. Receiver operating characteristic curve plot and the selected candidate parameters.** We obtained the five increased parameters of white blood cells, platelets, eosinophil, ALT, and CRP, as well as the five decreased parameters of red blood cell, hemoglobin, MCH, MCHC, and monocyte, between Kawasaki disease and febrile children.

also calculated using the ROC curve with Youden's index. The mean of c-index, sensitivity, specificity, and accuracy for training and the validation sets of five-fold cross-validation were 0.894, 0.766, 0.875, and 0.867 and 0.891, 0.755, 0.876, and 0.867, respectively. Finally, we found that a score of 14 had the best value of sensitivity plus specificity prediction rate for KD. The c-index, sensitivity, specificity, and accuracy for the testing set were 0.907, 0.824, 0.839, and

**Table 2. Area under the curve and cutoff points for candidate variables.**

|  | AUC (p-value) | Youden's index (cutoff point) | Candidate cutoff point |
|---|---|---|---|
| WBC (1000/µL) | 0.629 (<0.001) | 0.266 (9.35) | >10 |
| Platelets(1000/µL) | 0.692 (<0.001) | 0.298 (281.5) | >280 |
| Eosinophil (%) | 0.773 (<0.001) | 0.443 (1.35) | >1.5 |
| ALT (U/L) | 0.741 (<0.001) | 0.408 (29.5) | >30 |
| CRP (mg/L) | 0.721 (<0.001) | 0.403 (24.25) | >25 |
| RBC (million/µL) | 0.392 (<0.001) | 0.176 (4.425) | <4.5 |
| Hemoglobin (g/dL) | 0.306 (<0.001) | 0.290 (11.75) | <12 |
| MCH (pg/cell) | 0.376 (<0.001) | 0.198 (26.75) | <27 |
| MCHC (gHb/dL) | 0.345 (<0.001) | 0.219 (33.25) | <33 |
| Monocyte (%) | 0.274 (<0.001) | 0.359 (6.85) | <7 |

>indicates Kawasaki disease higher than controls; < indicates Kawasaki disease lower than controls.

**Table 3. Logistic regression analysis and final score system.**

| Variable | Univariate Est. (95%C.I.) | p-value | Multivariate Est. (95%C.I.) | p-value | Score |
|---|---|---|---|---|---|
| Gender (male) | 1.146 (0.910–1.443) | 0.248 | – | – | – |
| Age (year) | 0.961 (0.883–1.044) | 0.345 | – | – | – |
| WBC (1000/μL) | 2.993 (2.304–3.888) | <0.001 | – | 0.130 | – |
| Platelets>280 (1000/μL) | 3.506 (2.741–4.484) | <0.001 | 1.835 (1.363–2.470) | <0.001 | 2 |
| Eosinophil >1.5% | 6.673 (5.252–8.478) | <0.001 | 6.744 (5.040–9.023) | <0.001 | 7 |
| ALT>30 (U/L) | 7.299 (5.748–9.268) | <0.001 | 5.776 (4.325–7.712) | <0.001 | 6 |
| CRP>25 (mg/L) | 5.277 (4.048–6.881) | <0.001 | 5.878 (4.305–8.026) | <0.001 | 6 |
| RBC<4.5 (million/μL) | 1.905 (1.494–2.429) | <0.001 | – | 0.508 | – |
| Hemoglobin<12 (g/dL) | 3.138 (2.418–4.072) | <0.001 | 1.499 (1.090–2.060) | 0.013 | 1 |
| MCH<27 (pg/cell) | 2.116 (1.663–2.692) | <0.001 | 1.794 (1.328–2.424) | <0.001 | 2 |
| MCHC<33 (gHb/dL) | 2.614 (2.028–3.368) | <0.001 | 1.547 (1.104–2.168) | 0.011 | 2 |
| Monocyte <7 (%) | 4.569 (3.568–5.850) | <0.001 | 4.163 (3.125–5.545) | <0.001 | 4 |
| | | | | Total score | 30 |

0.838, respectively. Overall, the sensitivity, specificity, and accuracy for these 6,310 FC and 485 KD subjects were 0.798, 0.847, and 0.844, respectively (Table 4). The sensitivity of detecting complete KD and incomplete KD was 0.804 and 0.780, respectively.

As we know, positive predictive value (PPV) and negative predictive value (NPV) are sensitive to the proportion of the sample size (the proportion between the KD and FC groups, Table 5).Therefore, we decided to use another assessment criterion that was not affected by proportion, that is, likelihood ratio (LR), to evaluate the score system. Here, our score system has at least LR(+) = 5.22 (moderate evidence to rule in KD) and LR(-) = 0.24 (weak evidence to rule in FC) (Table 4).

## Verification of two independent KD cohorts

To test the performance of this scoring classification model, we enrolled two independent KD cohorts from two different institutes and different countries (103 KD patients from Kaohsiung Veterans General Hospital, Taiwan and 170 KD patients from Baoan Maternal and Child Health Hospital, Shenzhen, China). Given the specified parameters, the classification model had a 0.806 (83/103) and 0.765 (130/170) sensitivity, respectively. Altogether, it had a 0.780 (213/273) sensitivity, which is compatible with the performance of this model.

## Discussion

This study found that our score system in a large and diverse cohort of febrile children allowed for rapid discovery of patients with KD. Routine blood measurements of CBC/DC, CRP, and

**Table 4. Diagnostic performance with final score system in 6,310 FC and 485 KD.**

| | Sensitivity | Specificity | Accuracy | LR(+) | LR(-) |
|---|---|---|---|---|---|
| Training set | 0.766 | 0.875 | 0.867 | 5 | 0.26 |
| Validation set | 0.755 | 0.876 | 0.867 | 6.49 | 0.17 |
| Testing set | 0.824 | 0.839 | 0.838 | 5.11 | 0.21 |
| All | 0.798 | 0.847 | 0.844 | 5.22 | 0.24 |

LR: likelihood ratio.

**Table 5. Prevalence of KD and performance of score system.**

| Prevalence of KD | P(KD|+) | P(FC|-) |
|---|---|---|
| 485/6795[a] | 28.6% | 98.2% |
| 1/50 | 9.6% | 99.5% |
| 1/100 | 5.0% | 99.8% |
| 1/1,000 | 0.5% | 99.98% |

[a] indicates our patients; FC: febrile children; KD: Kawasaki disease.

+: Positive prediction by score system.

-: Negative prediction by score system.

other isolated parameters have a poor ability to differentiate KD from young FC. Of greater importance, without subjective clinical signs, our findings demonstrate that combining the values of platelets, eosinophil, ALT, CRP, hemoglobin, MCH, MCHC, and monocyte provides a good discriminatory ability for KD, while these parameters are also widely used in evaluating children with fever.

CRP levels have been previously recognized for their ability to identify or indicate febrile children with bacterial infection [17]. In our previous study of toll-like receptors in KD, we showed remarkable activation of toll-like receptor1, 2, 4, 5, 6, and 9, which is in line with a bacterial inflammatory response. Based on the results of this study, we also demonstrated that CRP levels are an important indicator for KD. In addition to standard clinical criteria [18], anemia is the most common clinical feature in KD patients and is thought to prolong the duration of active inflammation [19, 20]. Hemoglobin levels have also been shown to have the largest diagnostic differential between KD and FC [21] and to serve as a useful marker for differentiating KD shock syndrome from toxic shock syndrome in a pediatric intensive care unit [22]. In our study of these eight parameters, we also identified three parameters related to hemoglobin (Hb, MCH, MCHC) that decrease in KD patients. Hepcidin, which is encoded by the HAMP gene, is a well-known element of the inflammation-associated anemia mechanism [23]. In our previous studies, we have demonstrated that HAMP promoter hypomethylation upregulates hepcidin expression in KD patients [13, 24, 25]. The remarkable upregulation of hepcidin can induce transient anemia and hypoferremia in the acute inflammatory phase of KD [26], which is consistent with lower Hb, MCH, and MCHC in patients with KD.

KD patients experience a variety of non-specific clinical features, including gallbladder hydrops and impaired liver and function panel [26–30]. Various studies have shown that higher AST and ALT levels and lower albumin levels were correlated with IVIG treatment resistance and coronary artery lesion formation [31–33]. In this study, we discovered that ALT is more specific than AST in differentiating KD from febrile controls. In fact, increased gamma glutamyl transferase [30] and decreased albumin levels are common features in KD [34]; however; both items are not always collected from febrile children.

Eosinophils are potent effector cells implicated in allergic responses and helminth infections [35]. Eosinophils play the most important role in this score system, with the highest score of 7 out of a total score of 30. One recent study demonstrated that Z-score corrected eosinophil percentage was elevated in the acute phase of KD [36]. In addition to the acute phase, higher eosinophil percentage was also found in the convalescent phase following IVIG treatment [37]. From our previous reports, we not only found increased eosinophils in KD before IVIG treatment, but also discovered that it was even higher after IVIG treatment [38, 39]. Higher eosinophil levels in KD indicated a good treatment response of IVIG and lower CAL formation. The KD patients had higher eosinophil levels both before and after IVIG

therapy than the enterovirus patients, which was a T helper 2 immune response [40]. According to previous reports, eosinophils are a heterogeneous cell population and have different characteristics based on their site of residence. Taken altogether, the role of anti-inflammatory eosinophils as protective cells may play a more important role in KD than inflammatory eosinophils.

Monocytes also play an important role in this model. Persistent monocytosis after IVIG correlated with the development of CAL in KD. Lin et al. [41] observed augmented toll-like receptors (TLR) 2 expression on monocytes in both KD and a mouse model of coronary arteritis. Multiomics analyses identified epigenetic modulation of the S100A gene family in KD and their significant involvement in neutrophil transendothelial migration [42]. Armaroli et al. reported monocyte-derived interleukin-1β as the driver of S100A12-induced sterile inflammatory activation of human coronary artery endothelial cells in KD [43]. This finding highlights the interaction between monocytes and other cell populations in KD, as well as the importance of monocytes.

Neural network analysis has been used widely in medical fields to build predictive model. We have tried neural network analysis with the same setting: 83.3% of all cases for the training model and tuning the parameters of hidden layers and nodes. The neural network result had higher specificity (0.950) but much lower sensitivity (0.429) for KD diagnosis. In this study, we suggested a new score system without subjective clinical criteria in order to screen possible KD from FC. Neural network analysis is an effective predictive tool, but a score system may be easier to use in clinical settings.

IVIG resistance is another concern for pediatricians treating patients with KD. Therefore, we also tried to use our score model to predict IVIG resistance, but the accuracy was low. Although children with KD had higher score than FC (19.09 ± 6.29 vs. 8.41±5.30, p<0.001), there was no significant difference of scores between children with and without IVIG resistance (21.08 ± 7.12 vs. 19.15 ± 6.19, p = 0.205). This may be because our score system was not built up according to the treatment effect of IVIG and lacks some significant parameters that are important in predicting IVIG resistance, such as serum albumin and sodium levels [33, 44].

Early diagnosis of KD is required to ensure timely IVIG treatment and prevent adverse outcomes [5]. Ling et al. proposed a classification tool for differentiating KD from other febrile illnesses using both subjective clinical and objective laboratory test variables [10, 15]. However, first-line physicians in PED may not be so familiar with the subjective clinical manifestations of KD. On the other hand, since physicians know the clinical criteria of KD, they would be aware of the differential diagnosis of KD when evaluating children with fever in PED. In our findings, we proposed a novel scoring model with good ability to discriminate KD using only objective laboratory tests that are commonly used in PED. For clinical application, our novel scoring model may be integrated into the laboratory reporting system. When physicians check the laboratory test results, the application also provides the score to help first-line physicians be aware of the possibility of KD when evaluating children with fever in PED.

The current study has some limitations. First, we only included children aged under five years old to form our FC group. Application of the model in distinguishing KD from FC in children older than 5 years old warrants further study. Second, all the children in our modeling set, testing set, and validation set are Asian. Additional studies involving children of different countries and races are needed to assess the generalizability. Third, some other important laboratory tests, such as urinalysis, serum sodium and albumin levels, erythrocyte sedimentation, etc., which are commonly used in evaluating KD, were not included in our model because they were not routinely tested in our FC. However, these tests may play a crucial role in diagnosing KD.

## Conclusion

Our findings demonstrate a novel score system that combines ordinary laboratory data to achieve a good discriminatory ability for differentiating KD from other non-KD children with fever. The utilization of this novel score system can help physicians make early discovery of potential KD patients and may even prevent coronary artery lesions.

## Supporting information

**S1 Data.**
(PDF)

## Acknowledgments

We would like to thank the Clinical Trial Center, Kaohsiung Chang Gung Memorial Hospital, Kaohsiung, Taiwan for its statistical help.

## Author Contributions

**Conceptualization:** Ying-Hsien Huang, Ho-Chang Kuo.

**Data curation:** Chi-Hsiang Chu, Shih-Feng Liu, Ying-Hsien Huang.

**Formal analysis:** Chih-Min Tsai, Shih-Feng Liu, Ying-Hsien Huang.

**Funding acquisition:** Ho-Chang Kuo.

**Investigation:** Chih-Min Tsai, Chi-Hsiang Chu, Shih-Feng Liu, Ying-Hsien Huang.

**Methodology:** Chih-Min Tsai, Chi-Hsiang Chu, Xi Liu, Ken-Pen Weng.

**Project administration:** Ying-Hsien Huang.

**Resources:** Xi Liu, Ken-Pen Weng, Ho-Chang Kuo.

**Software:** Chi-Hsiang Chu.

**Supervision:** Ho-Chang Kuo.

**Validation:** Chih-Min Tsai, Chi-Hsiang Chu, Xi Liu, Ken-Pen Weng.

**Writing – original draft:** Chih-Min Tsai, Ying-Hsien Huang.

**Writing – review & editing:** Ho-Chang Kuo.

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
