## [Decision Letter · Decision Letter 0]

23 Sep 2020

PONE-D-20-24219

A novel score system of blood tests in prediction Kawasaki disease from febrile children

PLOS ONE

Dear Dr. Kuo,

Thank you for submitting your manuscript to PLOS ONE. After careful consideration, we feel that it has merit but does not fully meet PLOS ONE’s publication criteria as it currently stands. Therefore, we invite you to submit a revised version of the manuscript that addresses the points raised during the review process.

Specifically, both reviewers expressed concerns over how the manuscript defined the difference between complete and incomplete Kawasaki disease, and had several questions related to the analysis of the data. Please submit a revised manuscript with a rebuttal to each of the reviewers concerns by Nov 07 2020 11:59PM. If you will need more time than this to complete your revisions, please reply to this message or contact the journal office at plosone@plos.org.

We look forward to receiving your revised manuscript.

Kind regards,

Colin Johnson, Ph.D.

Academic Editor

PLOS ONE

Journal Requirements:

2. Please address the following:

- Please refer to any post-hoc corrections to correct for multiple comparisons during your statistical analyses. If these were not performed please justify the reasons. Please refer to our statistical reporting guidelines for assistance (https://journals.plos.org/plosone/s/submission-guidelines.#loc-statistical-reporting).

- Please ensure you have thoroughly discussed any potential limitations of this study within the Discussion section, including the potential impact of confounding factors.

- In your Methods section please include the dates upon which authors accessed the clinical data sources used in this study.

Reviewers' comments:

Reviewer's Responses to Questions

**Comments to the Author**

1. Is the manuscript technically sound, and do the data support the conclusions?

Reviewer #1: Yes

Reviewer #2: Yes

2. Has the statistical analysis been performed appropriately and rigorously? 

Reviewer #1: No

Reviewer #2: Yes

3. Have the authors made all data underlying the findings in their manuscript fully available?

Reviewer #1: Yes

Reviewer #2: Yes

4. Is the manuscript presented in an intelligible fashion and written in standard English?

Reviewer #1: Yes

Reviewer #2: Yes

5. Review Comments to the Author

Reviewer #1: The author(s) described a novel score system of blood tests to differentiate Kawasaki disease from febrile children. Although this paper might have some clinical impacts on medical staffs in the pediatric emergency departments, there are many questions to be solved in the technical and scientific points of view.

Technical comments:

1. To evaluate the goodness of statistical model in terms of prediction, c-index is more popular than accuracy. Thus, it is necessary to add the data of c-index in addition to sensitivity, specificity and accuracy throughout this paper.

2. In Table 1, the author described a lot of P values of P<0.001, which is basically due to the large number of patients in this study. Thus, it is necessary to evaluate the effect size such as Cohen’s d to select the useful parameters for the multivariate analyses.

3. In Table 2, the author used AUC of greater than 0.6 or less than 0.4. This seems to be too loose to evaluate the prediction ability of the statistical models. Usually, it should be greater than 0.7 or less than 0.3. Thus, it is necessary to correct Table 3 for selecting useful parameters. If necessary, the results using both of the criteria should be shown to discuss which is better for prediction of Kawasaki disease by using the model and verification samples. This might also contribute to simplify the score system presented in this paper.

4. What parameters in Table 3 were included in the multivariate logistic regression analysis? What about age and gender? They were essential for the analysis. I wonder if all other laboratory data were included in the multivariate analyses or not.

5. Recently, we often found the paper using the neural network analysis in the medical journals. Thus, the reviewer thought it necessary to add the neural network analysis in addition to the multivariate logistic regression analysis to compare the respective and also combined data.

Scientific Comments:

1. It is necessary to describe precisely the definitions for the incomplete Kawasaki disease in this paper. It is because all of the readers for PLOS ONE did not understand the difference between complete and incomplete Kawasaki disease.

2. It is well known that coronary artery lesions are sometimes observed in patients with incomplete Kawasaki disease, which suggested the importance of correct discrimination of incomplete Kawasaki disease from the control FC groups. In this paper, only the data of sensitivity was shown for the complete and incomplete Kawasaki disease respectively using the model data. Thus, it is preferable for the author to show the same analysis using verification data. This should contain the values of sensitivity, specificity, accuracy and c-index.

3. The reviewer also had a question whether the patient with suspected Kawasaki disease, who showed only no less than two sign and symptoms, were excluded correctly in the control FC groups.

4. Moreover, the reviewer also had a question if this model is applicable to discriminate IVIG resistance in patients with Kawasaki disease. It is because this is a very important point for doctors working in the pediatric emergency department.

5. When did the author(s) obtain the verification data? It is probably different from the model data of 2009 to 2017. It should be described in the section of study population.

Minor comments:

1. As the author indicated, this paper investigated only patients aged less than five years old. However, in the field of Kawasaki disease, incomplete cases aged more than 5 years old were often reported to have coronary artery lesions. Thus, the applicable age for the presented program should be described in the limitation of this paper.

2. Inappropriate usage of technical term was observed. For example, GPT in Table 3, while AST in the discussion. Probably, the latter is to be appropriate in the recent medical journal.

Reviewer #2: I uploaded my recommendations.

This is a very interesting paper, that if it could be reproduce in children from all races and countries, It could help practitioners from the Emergency Department who are not commonly aware of Kawasaki disease recognize this disease from other febrile illness and start an early IVIG treatment

6. PLOS authors have the option to publish the peer review history of their article (what does this mean?). If published, this will include your full peer review and any attached files.

Reviewer #1: No

Reviewer #2: No

---

## [Author Response · Author response to Decision Letter 0]

2 Nov 2020

Comments to the Author

Reviewer #1: The author(s) described a novel score system of blood tests to differentiate Kawasaki disease from febrile children. Although this paper might have some clinical impacts on medical staffs in the pediatric emergency departments, there are many questions to be solved in the technical and scientific points of view.

Technical comments:

1. To evaluate the goodness of statistical model in terms of prediction, c-index is more popular than accuracy. Thus, it is necessary to add the data of c-index in addition to sensitivity, specificity, and accuracy throughout this paper.

Reply: Thanks for your kindly reminder. We have added the c-index in our results. (In Table 1 and Line 188 – 192) 

2. In Table 1, the author described a lot of P values of P<0.001, which is basically due to the large number of patients in this study. Thus, it is necessary to evaluate the effect size such as Cohen’s d to select the useful parameters for the multivariate analyses.

Reply: Thanks for your kindly suggestion. We fully agree with your point of view and have added effect size in Table 1 and corrected some values. Please see Table 1.

3. In Table 2, the author used AUC of greater than 0.6 or less than 0.4. This seems to be too loose to evaluate the prediction ability of the statistical models. Usually, it should be greater than 0.7 or less than 0.3. Thus, it is necessary to correct Table 3 for selecting useful parameters. If necessary, the results using both of the criteria should be shown to discuss which is better for prediction of Kawasaki disease by using the model and verification samples. This might also contribute to simplify the score system presented in this paper.

Reply: Thanks for your kindly suggestion. We have used strict criterion (AUC greater than 0.7 or less than 0.3) to select the candidate markers before modeling, the performance of update model is not better than the original model. The mean of c-index, sensitivity, specificity, and accuracy for training and validation sets of five-fold cross-validation were 0.873, 0.723, 0.867, and 0.860 and 0.872, 0.720, 0.869, and 0.861, respectively. We prefer maintaining original criterion over using strict criterion. 

4. What parameters in Table 3 were included in the multivariate logistic regression analysis? What about age and gender? They were essential for the analysis. I wonder if all other laboratory data were included in the multivariate analyses or not.

Reply: Thanks for your kindly reminder. We have renewed the Table 3 to contain the result of age and gender. In our procedure of analysis, all candidate parameters from selecting stage were included in the multivariate logistic regression analysis. Please see Table 3.

5. Recently, we often found the paper using the neural network analysis in the medical journals. Thus, the reviewer thought it necessary to add the neural network analysis in addition to the multivariate logistic regression analysis to compare the respective and also combined data.

Reply: Thanks for your kindly suggestion. We apply the IBM SPSS “Neural Networks” for neural network analysis. Here we use the same setting: 83.3% of all cases for training model and tuning the parameters of hidden layer and nodes; 16.7% of all cases for testing the performance of out-of-sample. The best model is contained 1 hidden layer with 4 nodes. The sensitivity, specificity, and accuracy for training and testing sets were 0.494, 0.987, and 0.950 and 0.429, 0.985, and 0.958, respectively. This result had higher specificity (for febrile diagnosis) and accuracy (92.8% febrile children) but significantly low sensitivity for KD diagnosis (less than 50%). In this study, we suggested a new score system without subjective clinical criteria in order to screen possible KD from FC. Neural network analysis is an effective predictive tool, but a score system may be easier to use in clinical settings.

We have added some discussion about neural networks in the manuscript. Please refer to Line 272 – 277. 

Scientific Comments:

1. It is necessary to describe precisely the definitions for the incomplete Kawasaki disease in this paper. It is because all of the readers for PLOS ONE did not understand the difference between complete and incomplete Kawasaki disease.

Reply: Thanks for your kindly suggestion. We have added the definitions of complete/incomplete Kawasaki disease in manuscript. Please refer to Line 104 – 116.

2. It is well known that coronary artery lesions are sometimes observed in patients with incomplete Kawasaki disease, which suggested the importance of correct discrimination of incomplete Kawasaki disease from the control FC groups. In this paper, only the data of sensitivity was shown for the complete and incomplete Kawasaki disease respectively using the model data. Thus, it is preferable for the author to show the same analysis using verification data. This should contain the values of sensitivity, specificity, accuracy and c-index.

Reply: Thanks for your kindly reminder. We have added the c-index in the results. Please see lines 188 – 192.

3. The reviewer also had a question whether the patient with suspected Kawasaki disease, who showed only no less than two sign and symptoms, were excluded correctly in the control FC groups.

Reply: Thanks for your kindly reminder. We follow the guideline to diagnose KD and we excluded any suspected KD children from FC group. We have revised our manuscript in Line 119 – 122:

These controls presented to the PED of Kaohsiung Chang Gung Memorial Hospital, Taiwan with fever during the period of January 2009 through December 2015. Children suspected of having KD were excluded from the FC group. The clinical data sources of the control group were accessed by Prof. Y-H Huang and Dr. C-M Tsai.

4. Moreover, the reviewer also had a question if this model is applicable to discriminate IVIG resistance in patients with Kawasaki disease. It is because this is a very important point for doctors working in the pediatric emergency department.

Reply: We fully agree with the point of view in regarding discriminating IVIG resistance in patients with KD. However, the current model is focusing on helping physician to be aware of the differential diagnosis of KD. 

We added some discussion in our manuscript: (Line 278 – 284 )

Although children with KD had higher score than FC (19.09 ± 6.29 vs. 8.41±5.30, p<0.001), there was no significant difference of scores between children with and without IVIG resistance (21.08 ± 7.12 vs. 19.15 ± 6.19, p=0.205). This may be because our score system was not built up according to the treatment effect of IVIG and lacks some significant parameters that are important in predicting IVIG resistance, such as serum albumin and sodium levels.

5. When did the author(s) obtain the verification data? It is probably different from the model data of 2009 to 2017. It should be described in the section of study population. The same ? 

Reply: Thanks for your kindly reminder. We have added some information about these in our manuscript (Line 153 – 155):

We also used two other independent KD cohorts from different hospitals (Baoan Maternal and Child Health Hospital in China, N= 170, and Kaohsiung Veterans General Hospital in Taiwan, N= 103, data accessed by X Liu and K-P Weng, respectively, and with the same period of our KD group of 2009 to 2017) for final verification.

Minor comments:

1. As the author indicated, this paper investigated only patients aged less than five years old. However, in the field of Kawasaki disease, incomplete cases aged more than 5 years old were often reported to have coronary artery lesions. Thus, the applicable age for the presented program should be described in the limitation of this paper.

Reply: Thanks for your kindly opinion. We have added one paragraph about the limitation of our current study. (Line 296 – 303)

The current study has some limitations. First, we only included children aged under five years old to form our FC group. Application of the model in distinguishing KD from FC in children older than 5 years old warrants further study. Second, all the children in our modeling set, testing set, and validation set are Asian. Additional studies involving children of different races are needed to assess the generalizability. Third, some other important laboratory tests, such as urinalysis, serum sodium and albumin levels, erythrocyte sedimentation, etc., which are commonly used in evaluating KD, were not included in our model because they were not routinely tested in our FC. However, these tests may play a crucial role in diagnosing KD

2. Inappropriate usage of technical term was observed. For example, GPT in Table 3, while AST in the discussion. Probably, the latter is to be appropriate in the recent medical journal.

Reply: Thanks for your kindly remind, we have corrected the usage of technical term.

Reviewer #2: I uploaded my recommendations.

This is a very interesting paper, that if it could be reproduce in children from all races and countries, It could help practitioners from the Emergency Department who are not commonly aware of Kawasaki disease recognize this disease from other febrile illness and start an early IVIG treatment

Commentaries.

The objective of the study is to establish which routine laboratory results taken in the emergency room in children with fever could be helpful in differentiating Kawasaki disease from other febrile illness. 

Is well established that diagnosis of Kawasaki disease is based on clinical signs and symptoms and the laboratory results are more helpful when there are incomplete or atypical presentations of the disease. I think that it is very bold to try to make a diagnosis of Kawasaki disease without the use of clinical picture, although it will be very useful to add a tool to guide doctors in the emergency department think of Kawasaki disease.

Title

We all know that diagnosis of Kawasaki disease is based on clinical signs and symptoms, and the use of a construct based on routine laboratory results will be very helpful to diagnose the disease. The objective of the score is to select which patients are more likely to have Kawasaki disease based only on blood exams, so I think it is better to change “prediction” to “for differentiating” Kawasaki disease from febrile children. 

Reply: We agree with the reviewer’s opinion and revised our title as: A novel score system of blood tests for differentiating Kawasaki disease from febrile children

Abstract

As per journal guidelines, abbreviations are not recommended to be included in the abstract.

Reply: Thanks for your friendly reminder. We have revised our abstract and abbreviations are not included.

Introduction

It is important to establish the difference between complete and incomplete Kawasaki disease. I suggest that you correct the definition of classical or complete Kawasaki disease. (Lines 77 to 80) and make the difference of incomplete Kawasaki disease (line 80 “or fever lasting more than ….)

Reply: Thanks for your kind reminder. We have revised the definition of complete/incomplete Kawasaki disease in our manuscript. (Line 104 – 116):

The diagnosis of complete KD is based on the presence of ≥ 5 days of fever and the presence of ≥ 4 of the 5 principal clinical features. These five features consist of the following: (1) erythema and cracking of lips, strawberry tongue, and/or erythema of oral and pharyngeal mucosa, (2) bilateral bulbar conjunctival injection without exudate, (3) rash: maculopapular, diffuse erythroderma, or erythema multiforme-like, (4) erythema and edema of the hands and feet in the acute phase and/or periungual desquamation in the subacute phase, and (5) cervical lymphadenopathy (≥1.5 cm diameter), usually unilateral. In the presence of ≥ 4 principal clinical criteria, particularly when redness and swelling of the hands and feet are present, a KD diagnosis may be made with only 4 days of fever. Children with fever ≥ 5 days and 2 or 3 compatible clinical criteria are then evaluated with additional laboratory tests. If these children had ≥3 supplemental laboratory criteria (anemia for age, platelet count ≥450,000 after the seventh day of fever, albumin ≤3.0 g/dL, elevated ALT level, WBC count ≥15,000/mm3, and ≥10 WBC/HPF on urinalysis) or a positive echocardiogram, a diagnosis of incomplete KD was made.

Statistical Analysis

Because the authors already have made all the analysis, they have to know which statistical tests were used and why they used it. 

Reply: Thanks for your kind opinion.

Analysis process

I think this is the most important part of the paper, and according to the proposed methods it is accurate for evaluating the objectives of the study.

Reply: Thanks for your kind opinion.

Results

The number of patients included in the study is enough to make a good statistical analysis. 

It is of note that in the univariate analysis (which one they finally used?) all except but one of the laboratory results used to compare Kawasaki disease patients with febrile patients had statistically significant differences.

Reply: Thank your for allowing us to explain more about our univariate analysis. First, we used the ROC curve to select the candidate variables and determined their cut-off point in predicting KD. Only the parameters with significance of area under the curve (AUC) greater than 0.6 or less than 0.4 were selected for further analysis with logistic regression. Therefore, only 10 parameters listed on Table 2 were finally used for logistic regression analysis (shown as Table 3).

Also of note, is that some of laboratory parameters that they use to try to differentiate KD from other febrile illness (eosinophil percentage. MCH, MCHC and monocyte percentage) have not been fully recognized as important in Kawasaki disease. And more important that after the multivariate analysis, that a higher eosinophil percentage is useful in recognizing KD. A z-score corrected elevated eosinophil percentage has been reported in the acute phase of Kawasaki disease has been recently reported by Kwak JH, Lee JH, Ha KS. Significance of differential characteristics in infantile Kawasaki disease. Korean Circ J 2019;49:755-765. And also in the convalescent phase after IVIG treatment. (Tremoulet AH, Jain S, Chanraseakar D, Sun X, Sato Y, Burns JC. Evolution of Laboratory values in patients with Kawasaki disease. Pediatr Infect Dis J 2011;30:1022-1026). 

List in discussion. But we want a set of straightforward data. 

Reply: Thanks for your suggestion. We have revised and added these 2 references in our discussion to address the importance of eosinophil in children with KD. (Line 252 – 254) 

Discussion

I think it is appropriate and discuss all the important findings of the proposed model. 

Although the study has the limitation that it is performed only in children of Asian race, and it needs to be evaluated in children from other countries and races.

List in limitations.

Reply: We fully agree with your point of view. We have added this point to our limitations. (Line 298 – 300 ):

Second, all the children in our modeling set, testing set, and validation set are Asian. Additional studies involving children of different countries and races are needed to assess the generalizability.

References

The reference number 2 is incomplete

I suggest the author review the 2 references mentioned in the results review (Kwak and Tremoulet)

Reply: Thanks for your kind reminder. We have corrected the reference number 2 and added these 2 references. (references number 36 and 37)

This paper adds a new score to evaluate Kawasaki disease in the emergency room by doctors not used to diagnose the disease. It is worth publishing it with small corrections

Reply: Thanks for your friendly opinion.

Thank you.

---

## [Decision Letter · Decision Letter 1]

16 Dec 2020

A novel score system of blood tests for differentiating Kawasaki disease from febrile children

PONE-D-20-24219R1

Dear Dr. Kuo,

We’re pleased to inform you that your manuscript has been judged scientifically suitable for publication and will be formally accepted for publication once it meets all outstanding technical requirements.

Kind regards,

Colin Johnson, Ph.D.

Academic Editor

PLOS ONE

Additional Editor Comments (optional):

Reviewers' comments:

Reviewer's Responses to Questions

**Comments to the Author**

1. If the authors have adequately addressed your comments raised in a previous round of review and you feel that this manuscript is now acceptable for publication, you may indicate that here to bypass the “Comments to the Author” section, enter your conflict of interest statement in the “Confidential to Editor” section, and submit your "Accept" recommendation.

Reviewer #1: All comments have been addressed

Reviewer #2: All comments have been addressed

2. Is the manuscript technically sound, and do the data support the conclusions?

Reviewer #1: Yes

Reviewer #2: Yes

3. Has the statistical analysis been performed appropriately and rigorously? 

Reviewer #1: Yes

Reviewer #2: Yes

4. Have the authors made all data underlying the findings in their manuscript fully available?

Reviewer #1: Yes

Reviewer #2: Yes

5. Is the manuscript presented in an intelligible fashion and written in standard English?

Reviewer #1: Yes

Reviewer #2: Yes

6. Review Comments to the Author

Reviewer #1: (No Response)

Reviewer #2: (No Response)

7. PLOS authors have the option to publish the peer review history of their article (what does this mean?). If published, this will include your full peer review and any attached files.

Reviewer #1: No

Reviewer #2: No

---

## [Editor Report · Acceptance letter]

12 Jan 2021

PONE-D-20-24219R1 

A novel score system of blood tests for differentiating Kawasaki disease from febrile children 

Dear Dr. Kuo:

I'm pleased to inform you that your manuscript has been deemed suitable for publication in PLOS ONE. Congratulations! Your manuscript is now with our production department. 

Kind regards, 

on behalf of

Dr. Colin Johnson 

Academic Editor

PLOS ONE